# Assessment of knowledge and preventive behavior regarding rubella virus among females in Riyadh, Saudi Arabia: A cross-sectional study

Jehad A. Aldali ⓘ *

Department of Pathology, College of Medicine, Imam Mohammad Ibn Saud Islamic University (IMSIU), Riyadh, Saudi Arabia

* jaaldali@imamu.edu.sa

## Abstract

### Background

Rubella is an acute viral infection that is influenced by various factors, e.g., climate, dietary habits, and hygiene practices. This study evaluated the knowledge and preventive behaviors regarding Rubella infection among females in Riyadh, Saudi Arabia.

### Methods

In this cross-sectional study, an online Arabic and English survey was administered using Google Forms. The survey was distributed to female participants from March 25, 2024 to May 27, 2024. The first section of the online survey covered age, gender, education level, and marital status information. The second section assessed the participants' knowledge of rubella, including its transmission methods, symptoms, severity, concerns, and effects on pregnant females and their embryos. Finally, immunization status and prevention mechanisms were addressed in the third section.

### Results

The online survey was completed by a total of 448 female participants. The participants consisted of 53.8% young participants (ages 18–25) from a variety of educational backgrounds, with 62.3% of them being single. The Rubella virus was acknowledged by the majority of participants (57.6%). Specifically, 57.6% of the participants asserted that they were aware of the rubella infection. Nevertheless, 51.6% were unaware of the methods by which rubella is transmitted. Furthermore, 55.4% of the participants reported that they were unaware of the significant health risks associated with rubella. The fact that 59.8% of the participants were unaware of rubella complications raises several significant questions. Fever and skin rash were identified as the most frequently reported symptoms, accounting for 23.4% of all cases.

**Data availability statement:** All relevant data are within the article and its supporting information files.

**Funding:** This work was supported and funded by the Deanship of Scientific Research at Imam Mohammad Ibn Saud Islamic University (IMSIU) (grant number IMSIU-DDRSP2501).

**Competing interests:** The author has no conflict of interest to declare.

We examined the reasons for non-vaccination among the unvaccinated participants and discovered that 36.2% cited a lack of awareness. More than half of the participants (64.1%) consistently adhere to hygienic practices.

## Conclusion

The findings of the study indicates that females in Riyadh possess adequate knowledge concerning the rubella virus. When information is available, there exists a readiness to acquire knowledge and implement preventive measures. Increased education regarding the rubella virus may enhance individuals' willingness to engage and adopt proactive measures. Therefore, enhancing rubella prevention behaviors is crucial to mitigate the risk of infection, notwithstanding the advantages of general hygiene practices.

## Introduction

Rubella, also referred to as German measles, is a viral infection that is a member of the togaviridae family that can impact individuals of any age and gender [1–3], and transmission occurs through respiratory droplets and close contact. Congenital rubella can occur when pregnant females contract the rubella virus during the first trimester, and the virus can be transmitted to the fetus through the placenta, leading to malformations in various organs, e.g., the heart, eyes, and ears. In severe cases, it can even result in the death of newborns and infants [4–6].

Rubella is characterized by a distinctive red rash with small, raised lesions, mild fever, joint pain, myalgia, arthritis, headache, itching, fatigue, loss of appetite, slight crusting, and lymphadenopathy. In addition, encephalitis, splenomegaly, miscarriage, Guillain–Barré syndrome, and thrombocytopenic purpura are less frequent sequelae [7]. Rubella is one of several viruses associated with arthritis and other musculoskeletal diseases [8]. Over 70% of adult females who are infected develop arthritis [9], and if the infection occurs during the first few weeks of pregnancy up to 85% of newborns are born with a pattern of growth restriction and major birth defects, which is referred to as congenital rubella syndrome (CRS). During the early stages of pregnancy, rubella can cause a number of complications [10].

Rubella, a highly contagious viral infection, has shown a significant decline in incidence globally due to the introduction of vaccination programs. In the United States, the introduction of the rubella vaccine in 1969 led to a rapid decline in cases, with reported cases dropping from 57,686 in 1969 (58 cases per 100,000 population) to fewer than 1,000 annually by 1983 (<0.5 cases per 100,000 population)[11]. In 1995, the UK had one case of acute rubella, down from 70 in 1988 [12] Similarly, in Europe, robust immunization programs have drastically reduced rubella incidence, with only 729 cases reported between December 2016 and November 2017, most of which occurred in Poland [4]. However, rubella and congenital rubella syndrome (CRS) remain public health challenges in low- and middle-income countries, where the estimated CRS incidence in the late 1990s ranged from 44 to 275 cases per

100,000 live births [13]. This disparity highlights the critical role of vaccination in controlling the disease, especially in regions with limited healthcare infrastructure.

In the United States, rubella and CRS became nationally notifiable in 1966, and the highest number of reported rubella cases was 57,686 in 1969 (58 cases per 100,000 population). After vaccine licensure in 1969, rubella incidence decreased rapidly, with fewer than 1000 cases reported annually by 1983 (<0.5 cases per 100,000 population), and a moderate revival was observed in 1990 and 1991 [11]. In addition, the incidence of acute rubella in the United Kingdom decreased from 70 cases per year in 1988 to a single case in 1995. Only 729 European rubella cases were reported between December 2016 to November 2017, with 93 laboratory-confirmed cases, and most cases (77.3%) occurred in Poland. In poorer countries, rubella and CRS persist. CRS was estimated at 44–275 cases per 100,000 live births in these countries in the late 1990s. Typically, CRS causes various developmental anomalies, e.g., blindness and deafness, in poorer countries [7]. Globally, approximately 100,000 infants are born with CRS each year [14,15]. In most cases, the only time rubella can cause harm to a fetus is when the infection is acquired during the first 16 weeks of pregnancy. Generally, the severity of the observed abnormalities increases in proportion to the earlier onset of the lesions [16].

There is a strong healthcare system in Saudi Arabia that is dedicated to the eradication of communicable diseases, e.g., rubella, and the extensive immunization programs incorporated into the national public health policy are a testament to this commitment, which includes the measles-mumps-rubella (MMR) vaccine which is offered free of charge to all children and provides reinforced protection against these diseases [17]. Over the years, the impact of this program has been significant, and the incidence of rubella in Saudi Arabia has declined dramatically since the initiation of the vaccination program [17]. Before the rubella vaccine was introduced in Saudi Arabia as part of routine childhood vaccination in 1991, the prevalence of rubella seropositivity among women of childbearing age was within the range of 90–95% as a result of natural infection, as demonstrated by numerous studies [18]. Being one of the most common and serious congenital infections associated with stillbirth and birth defects, the World Health Organization has suggested that 95% vaccination coverage is required to prevent congenital rubella. However, recently, with anti-vaccination behaviors observed in many countries due to safety concerns, rubella cases have continued to appear [19,20]. In 2020, a study in western Saudi Arabia found 88.9% of women immune to rubella, while another found a susceptibility of 24%.2 [20,21]. In 2019, WHO reported 62 rubella cases in Saudi Arabia, and 4 cases of congenital rubella syndrome in 2017–2018 [19].

Rubella cases remain high in many low- and middle-income countries (LMICs) due to awareness, vaccination, and prevention gaps. Rubella has been reduced in Saudi Arabia due to the national immunization program. Females in Riyadh still have misconceptions about the rubella virus, its transmission, and vaccination, emphasizing the need for targeted educational and preventive interventions. Rubella elimination and maternal and child health depend on closing these gaps.

This study sought to evaluate the knowledge, attitudes, and preventive behaviors of women for a variety of reasons, despite the fact that there has been a globally and in Saudi Arabia a decline in rubella cases as a result of successful vaccination programs. In the first place, despite the decline, there are still gaps in awareness and misconceptions regarding rubella and its prevention, particularly among women of childbearing age, who are at the highest risk of transmitting the virus to their unborn children. Secondly, the knowledge and behavior of the population are essential for addressing any lingering gaps that could impede progress, as rubella elimination is heavily reliant on sustained vaccination coverage and informed public health practices. Finally, the absence of research on this particular subject in Saudi Arabia underscores the necessity of establishing a baseline of public awareness to inform targeted interventions and guarantee the preservation and enhancement of rubella control accomplishments.

To the best of our knowledge, limited research has been conducted on this specific topic in Saudi Arabia, making this study a valuable contribution to the existing body of knowledge. In Riyadh, Saudi Arabia, where the cultural and social landscape is diverse, understanding the knowledge and preventive behavior of females toward the rubella virus is crucial. Thus, in this study, we assessed the level of awareness and preventive practices among females in Riyadh to gain insights into the potential gaps in education and healthcare interventions

## 2. Participants and methods

### 2.1. Study design and settings

This cross-sectional study was conducted to evaluate the knowledge and preventive behavior regarding the rubella virus among females in Riyadh, Saudi Arabia. The study was conducted in the Department of Pathology at the College of Medicine, Imam Mohammad Ibn Saud Islamic University, Riyadh, Saudi Arabia.

### 2.2. Participants

The participants were recruited from the general female population in the Riyadh province of Saudi Arabia from March 25 to May 27, 2024. The study targeted females from various educational institutions in Riyadh. The research team visited various schools, universities, hospitals, and health institutes to collect the contact information of potential participants.

### 2.3. Sample size calculation

The study sample size was calculated using the "Raosoft.com" power calculator. In Riyadh, the sample size of 385 females was sufficient to achieve 95% confidence with a 5% margin of error; however, a total of 448 participants participated in the study.

### 2.4. Inclusion and exclusion criteria

Appropriate inclusion and exclusion criteria were considered to ensure that the study population was relevant and appropriate for answering the research questions while minimizing confounding factors that could affect the findings.

**Inclusion criteria:**

- Gender: Female participants only.

- Location: Residents of Riyadh, Saudi Arabia.

- Consent: Willingness to participate in the study.

- Language: Ability to understand and respond in the language(s) in which the survey or assessment is conducted (e.g., Arabic and English).

**Exclusion criteria**

- Nonresidents: Individuals not residing in Riyadh.

- Males: Male participants were excluded because the study is specific to females.

- Language barriers: Inability to understand the survey or assessment language, which could lead to misinterpretation and inaccurate responses.

- Incomplete surveys: Participants who did not complete the survey or assessment in full were excluded.

### 2.5. Development of the questionnaire

In this study, a self-administered English-language (translated to Arabic) web-based questionnaire was developed, validated, and circulated via a link survey to various social media and communication platforms, e.g., email and WhatsApp. The survey was distributed to the participants from March 20, 2024 to May 10, 2024. Initially, the survey was distributed to a pilot sample of five faculty members to countercheck the validity of the questionnaire and identify any technical concerns. We attempted to design a survey that was as succinct as possible. In addition, we explained that the information provided would be only utilized for research purposes. Note that the participants were not identified and had the right to withdraw

from the process at any stage. The questionnaire included an introductory paragraph explaining the nature and objectives of the study and the voluntary and anonymous nature of participation. The first section of the online survey focused on participant demographic data, including age, gender, and marital status. The second section focused on the participants' knowledge about rubella, specifically their comprehension of the disease, their perception of the modes of transmission of rubella, their level of concern regarding rubella, their perspectives on the seriousness of the virus, and their awareness of the impacts of the virus on pregnant females and their embryos. In addition, information about vaccinations, including the importance of vaccination, vaccination status, and willingness to recommend vaccination, was assessed through a series of questions. These questions were designed to capture the respondents' perceptions, beliefs, and feelings toward the disease and preventive measures. The third section focused on information about the symptoms caused by rubella and the participants' rubella immunization status, and the final section focused on preventive measures against rubella.

### 2.6. Ethical statement

The Institutional Review Board, College of Medicine, Imam Mohammad bin Saud Islamic University approved the study (Ref. 613–2024), which was validated on March 20, 2024.

## 3. Results

### 3.1. Demographic characteristics

Table 1 shows the demographic characteristics of the participants, including nationality, age, gender, education level, and marital status. As can be seen, Saudi nationals comprised the majority of the participants (92.9%). Young people between 18 and 25 years old comprised 53.8% of the participants, 20.3% of the participants were between 31 and 40, 11.6% were between 26 and 30, 41 and 50, and 3.6% were over 50 years old.

In addition, bachelor's students made up a large proportion of the participants (63.8%), and 14% and 14.1% of the participants had completed higher education (masters and doctoral levels, respectively). A small proportion of the respondents were high school graduates.

More than half of the participants were single (62.3%), 35.7% were married, 0.2% were widowed, and 1.8% were divorced.

**Table 1. Demographic characteristics.**

|  |  | Effective | Percentage |
|---|---|---|---|
| Nationality | Saudi | 416 | 92.9% |
|  | None Saudi | 32 | 7.1% |
| Gender | Female | 448 | 100% |
| Age | 18–25 | 241 | 53.8% |
|  | 26–30 | 52 | 11.6% |
|  | 31–40 | 91 | 20.3% |
|  | 41–50 | 48 | 10.7% |
|  | > 50 | 16 | 3.6% |
| Education qualification | Bachelors | 286 | 63.8% |
|  | Masters | 63 | 14.1% |
|  | PhD | 61 | 13.6% |
|  | School | 38 | 8.5% |
| Marital status | Divorced | 8 | 1.8% |
|  | Married | 160 | 35.7% |
|  | Single | 279 | 62.3% |
|  | Widowed | 1 | 0.2% |

### 3.2. Participant comprehension of rubella

Table 2 shows the level of knowledge of rubella among the participants. Most of the participants (57.6%) were aware of the virus; however, a substantial proportion (42.4%) were unfamiliar with rubella. Note that only 1.8% of the participants considered themselves highly knowledgeable about the rubella virus. The remaining participants were divided among those who felt neutral (27.2%), those who were unfamiliar (25.4%), and those who were very unfamiliar (30.8%) with rubella.

Regarding the transmission of rubella, a large proportion (51.6%) of the participants lacked knowledge about rubella transmission routes. Among those with knowledge, most attributed transmission to direct transmission from an infected person. A minority of the participants (4.5%) believed that the virus was transmitted through airborne droplets, and 3.1% believed that it can be transmitted through contaminated surfaces. In contrast, 19.9% of the participants believed that the

**Table 2. Participant understanding of rubella.**

|  |  | Effective | Percentage |
|---|---|---|---|
| Heard about rubella before | No | 190 | 42.4% |
|  | Yes | 258 | 57.6% |
| The statement that best describes your knowledge of rubella | Neutral | 122 | 27.2% |
|  | Somewhat knowledgeable | 66 | 14.7% |
|  | Somewhat unfamiliar | 114 | 25.4% |
|  | Very knowledgeable | 8 | 1.8% |
|  | Very unfamiliar | 138 | 30.8% |
| Rubella transmission | Airborne droplets | 20 | 4.5% |
|  | Contaminated surfaces | 14 | 3.1% |
|  | Direct contact with an infected person | 94 | 21.0% |
|  | All of the above | 89 | 19.9% |
|  | I don't know | 231 | 51.6% |
| Concerned about the spread of rubella in the community | Concerned | 62 | 13.8% |
|  | Neutral | 163 | 36.4% |
|  | Not concerned at all | 85 | 19.0% |
|  | Not very concerned | 111 | 24.8% |
|  | Very concerned | 27 | 6.0% |
| Believe that rubella is a serious health threat | No | 71 | 15.8% |
|  | Unsure | 248 | 55.4% |
|  | Yes | 129 | 28.8% |
| Risk of infection with the rubella virus | Likely | 27 | 6.0% |
|  | Neutral | 131 | 29.2% |
|  | Unlikely | 181 | 40.4% |
|  | Very likely | 6 | 1.3% |
|  | Very unlikely | 103 | 23.0% |
| Knowledge of the potential complications of rubella, especially for pregnant women and their unborn children | No | 268 | 59.8% |
|  | Yes | 82 | 18.3% |
|  | Unsure | 98 | 21.9% |

virus can be transmitted through all three methods. Most of the participants were distributed between neutral (36.4%), unconcerned (19.0%), and not worried (24.8%) about the spread of rubella in their community. Only a small percentage of the participants (6.0%) were very concerned, and 6.0% were concerned about the spread of rubella in their community.

Regarding the severity of the health threat, 55.4% of the participants indicated they were uncertain or unsure about the seriousness of the rubella virus. This suggests that a significant portion of the participants may have lacked clarity or confidence in their understanding of the health risks, whereas 28.8% agreed with the statement and 15.8% disagreed. In terms of the likelihood of becoming infected with the rubella virus, 6.0% and 1.3% of the participants indicated that they were likely and very likely to become infected, respectively. In contrast, most of the participants were divided between neutral (29.2%), unlikely (40.4%), and very unlikely (23.0%).

Regarding complications due to rubella infection, it is concerning to note that 59.8% of the participants were unaware of the possible complications of rubella, 21.9% were unsure, and only 18.3% were aware of the complications. This is particularly alarming when it comes to pregnant females and their unborn children.

### 3.3. Signs and symptoms of rubella

As shown in Table 3, 105 participants (23.4%) stated that the most common symptoms reported were fever and skin rash simultaneously. The next most common symptoms were reported to be fever alone (61 participants; 13.6%) and skin rash

**Table 3. Rubella signs and symptoms.**

|  |  | Effective | Percentage |
|---|---|---|---|
| Symptoms of rubella | Cough | 5 | 1.1% |
|  | Fever | 61 | 13.6% |
|  | Fever, cough | 7 | 1.6% |
|  | Fever, headache | 12 | 2.7% |
|  | Fever, headache, cough | 17 | 3.8% |
|  | Fever, joint pain | 12 | 2.7% |
|  | Fever, joint pain, cough | 5 | 1.1% |
|  | Fever, joint pain, headache | 13 | 2.9% |
|  | Fever, joint pain, headache, cough | 2 | 0.4% |
|  | Fever, skin rash | 105 | 23.4% |
|  | C | 9 | 2.0% |
|  | H | 32 | 7.1% |
|  | Fever, skin rash, headache, cough | 8 | 1.8% |
|  | Fever, skin rash, joint pain | 30 | 6.7% |
|  | Fever, skin rash, joint pain, cough | 3 | 0.7% |
|  | Fever, skin rash, joint pain, headache | 19 | 4.2% |
|  | Fever, skin rash, joint pain, headache, cough | 23 | 5.1% |
|  | Headache | 1 | 0.2% |
|  | Joint pain | 5 | 1.1% |
|  | Joint pain, headache | 1 | 0.2% |
|  | Skin rash | 64 | 14.3% |
|  | Skin rash, cough | 1 | 0.2% |
|  | Skin rash, headache | 1 | 0.2% |
|  | Skin rash, joint pain | 8 | 1.8% |
|  | Skin rash, joint pain, cough | 1 | 0.2% |
|  | Skin rash, joint pain | 3 | 0.7% |

alone (46 participants; 14.6%), and 5.1% of the participants reported that all three symptoms occur simultaneously. Fever, skin rash, and joint pain were reported by 30 participants (6.7%). Finally, 7.1% of the participants reported experiencing all three symptoms simultaneously, and 23 participants (5.1%) reported that fever, skin rash, joint pain, cough and headache all occur simultaneously.

### 3.4. Status against Rubella

Table 4 shows that most participants (92.4%) had not received the rubella vaccination, and only a small percentage of the participants (7.6%) had received the vaccination.

The rubella vaccination status was uncertain for 48.4% of the participants, and 20.5% have been vaccinated, 6.3% have been vaccinated but are not current, and 18.9% are not and do not wish to be vaccinated.

Among those who had not been vaccinated, we explored their reasons and found that 36.2% of the participants cited a lack of awareness, 8% were concerned about the safety of the vaccine, and a small percentage lacked access to health-care (3.1%).

### 3.5. Preventative measures against rubella

In terms of rubella prevention strategies, Table 5 shows that 64.1% of the participants always take hygiene measures to prevent infection, while only 24.8% do so frequently, which is an alarming percentage that cannot be ignored. Relative to preventative measures, a small percentage of the participants take them occasionally (9.4%), and a negligible percentage of the participants do so infrequently (1.1%) or never (0.7%).

Approximately half of the participants stated that they were very likely (25.0%) or likely (27.9%) to encourage those around them to get vaccinated against rubella, and the remaining participants were distributed between neutral (33%) and not encouraging vaccination (6.3% unlikely and 7.8% very unlikely).

**Table 4. History of vaccination against rubella.**

|  |  | Effective | Percentage |
|---|---|---|---|
| Discussed the rubella vaccination with your healthcare provider during pregnancy or before planning to conceive | No | 414 | 92.4% |
|  | Yes | 34 | 7.6% |
| Received the rubella vaccine (MMR vaccine) | I am not sure | 217 | 48.4% |
|  | No, and I don't plan to get vaccinated | 87 | 19.4% |
|  | No, but I plan to get vaccinated | 28 | 6.3% |
|  | Yes, and I am up to date | 24 | 5.4% |
|  | Yes, but I am not up to date | 92 | 20.5% |
| If you have not been vaccinated, what is the main reason? | Concerns about vaccine safety | 36 | 8.0% |
|  | Lack of awareness | 162 | 36.2% |
|  | No access to healthcare | 14 | 3.1% |
|  | Other | 234 | 52.2% |
|  | Religious or personal beliefs | 2 | 0.4% |

**Table 5. Strategies for preventing rubella.**

| | | Effective | Percentage |
|---|---|---|---|
| Practice good hygiene habits, e.g., frequent handwashing, to prevent the spread of infectious diseases | Always | 287 | 64.1% |
| | Never | 3 | 0.7% |
| | Occasionally | 42 | 9.4% |
| | Often | 111 | 24.8% |
| | Rarely | 5 | 1.1% |
| Encourage others to get vaccinated against rubella | Likely | 125 | 27.9% |
| | Neutral | 148 | 33.0% |
| | Unlikely | 28 | 6.3% |
| | Very likely | 112 | 25.0% |
| | Very unlikely | 35 | 7,8% |

## 4. Discussion

The rubella virus causes an acute infectious disease among children and adults, characterized by fever, rash, and lymph-adenopathy. It is spread through respiratory droplets and close contact. The rubella virus typically causes lymphocytosis. However, the relevant symptoms may be mild, which makes disease surveillance difficult. In addition, infection of the rubella virus in pregnant females during the first three to four months of pregnancy can cause CRS, which results in heart, eye, ear, and organ malformations, and even death in neonates and infants [22].

This study has performed a thorough examination of the preventive behaviors and knowledge about rubella among female students in Riyadh, Saudi Arabia. The demographic analysis indicated that most participants were young Saudi females, with the majority being single between the ages of 18–25. Furthermore, the educational backgrounds of the participants were diverse, with most participants being bachelor's students, a substantial number holding masters or doctoral degrees, and a minority having completed only high school.

The results indicate that a significant number of the participants are unfamiliar with the rubella virus, even though more than 50% are aware of it to some degree. A similar study conducted in Brazil reported that most of the adult participants (69.9%) were aware of rubella; however, their understanding of the disease was inadequate [23]. According to the findings of another similar study conducted in India, 75% of the surveyed students not in the medical field were unaware of rubella [24].

A large percentage of the population in Riyadh, Saudi Arabia does not know how rubella is transmitted. While many attribute transmission to direct contact, only a few recognize airborne droplets or contaminated surfaces as potential routes for transmission. In addition, there is a wide range of concerns regarding rubella, with some individuals expressing a high level of concern about the spread of the disease in the community. Similar to the findings reported by AL-ABD et al. [25], the participants' comprehension of rubella transmission was relatively low, with nearly half being aware of the disease's transmission method. This is consistent with the results of the study conducted by Olajide OM et al., (22) which indicated that only a small number of respondents were aware of rubella and its transmission mechanisms. In that study, approximately 40% of the surveyed students identified breathing as a mode of transmission. Another study found that less than 20% of the respondents were aware of the different modes of transmission, which is in contrast to the findings of this study [26].

Regarding the potential harm that can be caused by rubella, more than half of the participants were uncertain about the severity of the danger rubella poses to their health, and only a small percentage believed they were likely to become infected with the disease. Rubella is characterized by numerous symptoms, the easiest to recognize being fever and a rash on the skin. However, a large percentage of the population is unaware of the potential complications that rubella can

cause, especially for pregnant women and their fetuses. This is consistent with the previous findings reported by AL-ABD et al. [25], who found that participants had limited knowledge about rubella, including its symptoms, transmission methods, and sources of infection. This aligns with the findings of previous studies in other countries, e.g., Egypt, that identified low awareness of rubella among females [27].

The assessment of vaccinations is integral to understanding the preventive behaviors and potential barriers among females in Riyadh (Lack of disease knowledge, cultural or religious misconceptions, limited healthcare access, vaccine side effects, or misinformation about vaccination may be barriers. Addressing these barriers can inform targeted public health interventions to boost vaccination rates and awareness). The insights gained from this study can inform targeted public health interventions to improve awareness and vaccination uptake. areas of concern. A substantial proportion of the participants were uncertain about whether they had received vaccinations, and a considerable number had not yet received vaccinations and indicated that they have no intention to become vaccinated. A lack of awareness, concerns about vaccine safety, and limited access to health care are the primary reasons for not being vaccinated. Supporting our findings, it was previously reported that between June 17 and August 12, 2014, rubella was most prevalent in Gokwe North District primary school children aged 5–9. Schoolchildren then spread the virus to the greater community. This outbreak was contained by school and home screening and isolation measures. In addition, health workers and communities should receive rubella training, and routine rubella vaccinations may prevent future outbreaks [28].

Taking preventive measures, e.g., maintaining sufficient hygiene, is a common practice, and most participants take these measures consistently. Nearly half of the survey respondents were likely to support vaccination against rubella, indicating a relatively high level of encouragement for vaccination among their peers.

Factors affecting the knowledge and preventive behavior regarding the rubella virus among females in Riyadh include education level, access to reliable health information, and healthcare access. Higher levels of education and better socioeconomic status are generally associated with greater awareness and understanding of rubella. Furthermore, cultural beliefs and social norms significantly influence knowledge about the disease and vaccination, while trust in the healthcare system and previous health experiences shape individual perceptions. Accessibility to vaccination services and exposure to health campaigns also enhance preventive behaviors. Personal health practices and the perceived severity and susceptibility to rubella also play important roles in determining and taking actions toward preventive measures.

By using community health seminars, integrating rubella education into routine health care visits, and leveraging social media campaigns to address misconceptions, health care providers have the potential to raise public awareness about rubella transmission and its complications. By working with religious and community leaders, it is possible to combat cultural myths. In addition, implementing awareness programs in schools and universities ensures that students are exposed to accurate information early on. Another benefit of providing health professionals with training in effective communication strategies is that it can improve patient education and encourage vaccination of the population. Collectively, these initiatives have the potential to improve public understanding and encourage more people to receive vaccinations.

## Limitations

We acknowledge that relying on self-reported data may introduce biases, e.g., social desirability bias, where participants provide responses that they perceive to be more socially acceptable or preferable. This can affect the accuracy of rubella-related behaviors and knowledge. This study revealed significant gaps in the participants' understanding of rubella; however, the depth of knowledge assessed may be limited by the design of the survey. More comprehensive or nuanced questions may reveal different levels of understanding and misconceptions about the disease. In addition, uncertainty about vaccination status among the participants highlights potential recall bias or a lack of accurate medical records. This uncertainty complicates the assessment of vaccination uptake and its association with preventive knowledge and behaviors. Furthermore, this study may not fully capture the influence of cultural, social, and familial factors on the prevalence

of rubella and vaccination, and these factors can play a crucial role in shaping health behaviors and perceptions, especially in a culturally diverse society like Saudi Arabia. Limited access to healthcare has been identified as a barrier to vaccination; however, this study did not consider specific barriers, e.g., geographic, economic, or systemic issues in the healthcare system. Thus, a more detailed investigation can provide clearer insights into how to address these barriers. The cross-sectional nature of this study provides a snapshot of knowledge at a single point in time. Therefore, additional longitudinal studies can provide further insight into how these aspects develop over time and in response to public health interventions or changing circumstances. The survey conducted in this study covered a range of topics, some areas, e.g., the specific sources of the participants' knowledge about rubella or their trust in different sources of information, were not explored in depth, and understanding these issues can help design more effective educational campaigns. Thus, future research should address these limitations to better understand rubella-related knowledge and behaviors among diverse Saudi populations to enable more targeted and effective public health strategies.

## Conclusion

The findings of this study into knowledge and practices regarding rubella among females in Riyadh, Saudi Arabia have indicated significant gaps and mixed levels of awareness. While more than half of the study population is aware of rubella, many are unfamiliar with its transmission methods, e.g., airborne droplets and contaminated surfaces, frequently associating it with only direct contact. In addition, concerns about the disease vary, with some participants showing high concern for community spread; however, most of the participants expressed uncertainty about the seriousness of rubella's health risks, and only a few considered themselves likely to contract the virus. Awareness of the virus's complications, particularly for pregnant women and their fetuses, was found to be low. Furthermore, vaccination awareness and uptake are concerning, and a notable portion of the participants expressed uncertainty about their vaccination status or expressed an unwillingness to get vaccinated due to safety concerns and limited access to healthcare. Despite this, maintaining hygiene is a common preventive practice, and nearly 50% of the participants support vaccination, showing a generally positive attitude toward immunization within the community.

## Supporting information

**S1 Data. This supporting material contains the complete survey instrument used in the current study.** It was designed to assess knowledge levels and preventive behaviors related to the rubella virus among females in Riyadh, Saudi Arabia. The questionnaire may be useful for other researchers conducting similar studies or exploring related public health topics. It is provided here to promote transparency, replicability, and potential adaptation in future research projects.
(XLSX)

## Author contributions

**Data curation:** JEHAD A ALDALI.

**Formal analysis:** JEHAD A ALDALI.

**Investigation:** JEHAD A ALDALI.

**Methodology:** JEHAD A ALDALI.

**Project administration:** JEHAD A ALDALI.

**Supervision:** JEHAD A ALDALI.

**Writing – original draft:** JEHAD A ALDALI.

**Writing – review & editing:** JEHAD A ALDALI.

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
