## [Decision Letter · Decision Letter 0]

26 Jun 2024

PONE-D-24-23371Assessment of Knowledge, Attitude, and Preventive Behavior Regarding Rubella Virus Among Females in Riyadh, Saudi ArabiaPLOS ONE

Dear Dr. ALDALI,

Thank you for submitting your manuscript to PLOS ONE. After careful consideration, we feel that it has merit but does not fully meet PLOS ONE’s publication criteria as it currently stands. Therefore, we invite you to submit a revised version of the manuscript that addresses the points raised during the review process.

We look forward to receiving your revised manuscript.

Kind regards,

Omar Enzo Santangelo

Academic Editor

PLOS ONE

Journal Requirements:

**Additional Editor Comments:**

Dear authors, the manuscript needs major revisions, please respond point by point to the reviewers' requests.

Kind regards

Reviewers' comments:

Reviewer's Responses to Questions

**Comments to the Author**

1. Is the manuscript technically sound, and do the data support the conclusions?

Reviewer #1: Yes

Reviewer #2: No

2. Has the statistical analysis been performed appropriately and rigorously? 

Reviewer #1: Yes

Reviewer #2: No

3. Have the authors made all data underlying the findings in their manuscript fully available?

Reviewer #1: Yes

Reviewer #2: No

4. Is the manuscript presented in an intelligible fashion and written in standard English?

Reviewer #1: Yes

Reviewer #2: Yes

5. Review Comments to the Author

Reviewer #1: Excellent work, but please review some grammers and some words with capita letters make them with small letters. In the metod section there is wrong spelling in the month. Please add inclusion and exclusion criteria of the participants.

Reviewer #2: The following are important points for the manuscript.

1. Abstract:

1.1. In abstract, the method section is poor, it is not well written. It lacks important data such as statistical analysis, study variables, and population.

1.2. The result section in abstract is deficient and very poor, it is not supported by enough findings. It was written as if it is conclusion, no values or interpretations found.

1.3. The authors mentioned in result section that (Awareness of rubella's symptoms, like fever and rash, is common) and (Vaccination awareness and uptake are low).These findings are not supported by number or percentage.

1.4. In abstract, conclusion section is poor, it does not represent the findings.

2. Introduction: suitable and expressive.

The objectives are clear, and so the presentation

3. Methods : the methods are well described. The deficient point that the authors did not mention how the attitude is assessed. Under the section 2.4. Study Questionnaire Development, the authors described various parts of the survey without writing anything about the attitude although it was a main part in the title of the manuscript.

4. Results section:

This part is poor. The analysis is not satisfied. The authors did not write the assessment of attitude among the participants. I have observed the following deficient points:

4.1. The authors did not obtain the score levels of knowledge and attitude

4.2. The authors did not make comparisons among the participants such as males and females, marital status, education levels, and so on

4.3. The authors did mention the factors affecting the Knowledge, Attitude, and Preventive Behavior Regarding Rubella Virus.

I suggest the followings:

4.4. The authors should obtain score levels for Knowledge, Attitude, and Preventive Behavior, and compare the mean/median levels among the participants using chi-square test

4.5. The authors should add a regression table to demonstrate the factors that influence the Knowledge, Attitude, and Preventive Behavior Regarding Rubella Virus

6. PLOS authors have the option to publish the peer review history of their article (what does this mean? ). If published, this will include your full peer review and any attached files.

**Do you want your identity to be public for this peer review?** For information about this choice, including consent withdrawal, please see our Privacy Policy .

Reviewer #1: **Yes: ** Yosra Alhindi

Reviewer #2: No

---

## [Author Response · Author response to Decision Letter 1]

3 Aug 2024

Thank you for your thoughtful feedback and for recognizing the significance of our research on the Rubella virus, specifically concerning the Assessment of Knowledge, Attitude, and Preventive Behavior Regarding the Rubella Virus Among Females in Riyadh, Saudi Arabia. We appreciate your interest in our study and your constructive comments.

We acknowledge your suggestions regarding the manuscript's reporting and agree that improvements can be made to enhance clarity and comprehensiveness. We are committed to refining the manuscript to better present our findings and ensure that the information is accessible and valuable to the readers.

We carefully reviewed your specific points and incorporated the necessary revisions to address the areas of concern. This will include a more detailed explanation of our methodology, a clearer presentation of our results, and a more in-depth discussion of the implications of our findings.

Thank you once again for your valuable input. We are resubmitting a revised version that meets the high standards of the journal and contributes meaningfully to the field of public health research.

---

## [Decision Letter · Decision Letter 1]

2 Sep 2024

PONE-D-24-23371R1Assessment of Knowledge, Attitude, and Preventive Behavior Regarding Rubella Virus Among Females in Riyadh, Saudi ArabiaPLOS ONE

Dear Dr. ALDALI,

Thank you for submitting your manuscript to PLOS ONE. After careful consideration, we feel that it has merit but does not fully meet PLOS ONE’s publication criteria as it currently stands. Therefore, we invite you to submit a revised version of the manuscript that addresses the points raised during the review process.

**ACADEMIC EDITOR: **

Dear authors, the manuscript requires major revisions, please respond point by point to the reviewers' indications. 

Kind regards.

We look forward to receiving your revised manuscript.

Kind regards,

Omar Enzo Santangelo

Academic Editor

PLOS ONE

Reviewers' comments:

Reviewer's Responses to Questions

**Comments to the Author**

1. If the authors have adequately addressed your comments raised in a previous round of review and you feel that this manuscript is now acceptable for publication, you may indicate that here to bypass the “Comments to the Author” section, enter your conflict of interest statement in the “Confidential to Editor” section, and submit your "Accept" recommendation.

Reviewer #1: All comments have been addressed

Reviewer #2: (No Response)

Reviewer #3: (No Response)

2. Is the manuscript technically sound, and do the data support the conclusions?

Reviewer #1: Yes

Reviewer #2: Yes

Reviewer #3: Partly

3. Has the statistical analysis been performed appropriately and rigorously? 

Reviewer #1: Yes

Reviewer #2: No

Reviewer #3: No

4. Have the authors made all data underlying the findings in their manuscript fully available?

Reviewer #1: Yes

Reviewer #2: Yes

Reviewer #3: No

5. Is the manuscript presented in an intelligible fashion and written in standard English?

Reviewer #1: Yes

Reviewer #2: Yes

Reviewer #3: No

6. Review Comments to the Author

Reviewer #1: (No Response)

Reviewer #2: 1. Abstract:

1.1. In abstract, the method section is poor, it is not well written. It lacks important data such as statistical analysis, study variables, and population.

1.2. The result section in abstract is deficient and very poor, it is not supported by enough findings. It was written as if it is conclusion, no values or interpretations found.

1.3. The authors mentioned in result section that (Awareness of rubella's symptoms, like fever and rash, is common) and (Vaccination awareness and uptake are low).These findings are not supported by number or percentage.

1.4. In abstract, conclusion section is poor, it does not represent the findings.

2. Introduction: suitable and expressive.

The objectives are clear, and so the presentation

3. Methods : the methods are well described. The deficient point that the authors did not mention how the attitude is assessed. Under the section 2.4. Study Questionnaire Development, the authors described various parts of the survey without writing anything about the attitude although it was a main part in the title of the manuscript.

4. Results section:

This part is poor. The analysis is not satisfied. The authors did not write the assessment of attitude among the participants. The results are just percentage without interpretations. I have observed the following deficient points:

4.1. The authors did not obtain the score levels of knowledge and attitude

4.2. The authors did not make comparisons among the participants such as males and females, marital status, education levels, and so on

4.3. The authors did mention the factors affecting the Knowledge, Attitude, and Preventive Behavior Regarding Rubella Virus.

I suggest the followings:

4.4. The authors should obtain score levels for Knowledge, Attitude, and Preventive Behavior, and compare the mean/median levels among the participants using chi-square test

4.5. The authors should add a regression table to demonstrate the factors that influence the Knowledge, Attitude, and Preventive Behavior Regarding Rubella Virus

Reviewer #3: 1) Many grammar and editing errors 

 rubella --- r should be capital in the abstract and whole manuscript, but many are unclear about how it spreads.

2) The objectives should be smart (person, time, place) in the abstract and the whole manuscript (add the time).

3) in the abstract---------- The study included a total of 448 participants, all of whom were females (replace to be) (of 448 females (aged -------)

4) In the abstract, An online survey using Google Forms in English and translated into Arabic was used to record information----- an online Arabic survey using Google Forms --- no need to this detail in the abstract 

5) The term gender is a behavioral term, not a physiological term, so it replaces sex in the whole manuscript. 

6)  The combination of fever and skin rash was the most frequently reported symptom, accounting for 23.4% of all cases. ----- cases of rubella ----- Add a number before the %.

7)  Regarding the serious health risk, 55.4% of the participants were unsure of the situation.---- What does it mean I don't know or 

8)  in certain areas

 ---- females, women, cases, participants fix the term in the whole manuscript ---- to be females as in the title 

------ in the Kingdom of Saudi Arabia and  Saudi Arabia-----fix the term in the whole manuscript to be as in the title. 

-----This study aims to assess the level of awareness, attitudes,------- knowledge in others ---fix the term in the whole manuscript to be as in the title

9)  The KAP survey , US  -- add the full name for the first time 

10) in the abstract --- . Questions were included in the questionnaire to evaluate awareness of rubella, knowledge of symptoms, transmission, and preventative measures of rubella infection--------- where is the attitude 

and the awarness is part of the knowledge 

11) you discuss the global incidence in numbers in details ----- and The impact of this program over the years has been significant. The incidence of Rubella in Saudi Arabia has dramatically declined since the initiation of 

the vaccination program (17).------------------------ kindly add the numbers in detail 

12) add a brief about the   initiation of the vaccination program --- in Saudi Arabia (started when, achievement)---in the introduction 

13) what is your rational ---In Saudi Arabia, limited research has been conducted on this specific topic, making this 

study a valuable contribution to the existing body of knowledge--- add  reference .

14)  in the Department of Pathology, collage of Medicine, Imam Mohammad Ibn Saud Islamic University, Riyadh, Saudi Arabia.------- why the department of pathology ??? 

15) In the Riyadh province of Saudi Arabia regional the sample size of 385 participants was 

sufficient to achieve 95% confidence with a 5% margin of error.----- as a  cross sectional study you need a prevlance or incidence (with refrence) --- and the total popoulation or females in Riyadh 

16)The second section focused on the 

information the Knowledge about rubella, specifically interested in their 

comprehension of the disease, their perception of the modes of transmission of rubella, 

their level of concern regarding it, their perspectives on the gravity of the virus, and 

their awareness of the impact of this virus on pregnant women and their embryos. The 

The third section focused on information about the symptoms caused by rubella. 

immunization status.  The last section focuses on preventive measures against rubella. -----  add refrence ????

17) The third section focused on information about the symptoms caused by rubella—its knowledge q 

 Rubella immunization status----- It is a part of preventive measures (rewrite).

19) while women comprise 96.2%.--- what does it mean

20) the age groups should be continous , and in order in table 1

25-<30

30-<40

40-<50

under 18-<25

years > 50

why the age intervals is not fixed ( 7 y then 4-year -10 year)

21) in all tables should be f not Effective

22) the symptoms should be calculated for each symptoms not merged

23) in knowledge study ---- should be total knowledge score

where does it then classify groups s that a fair number of participants are known of the rubella virus, but many are unclear about how it spreads---- what is fair mean

24) Table 5: Strategies for preventing rubella ---- should be the practice of preventive measure including vaccination status

11) where is the attitude tables ?

7. PLOS authors have the option to publish the peer review history of their article (what does this mean? ). If published, this will include your full peer review and any attached files.

**Do you want your identity to be public for this peer review?** For information about this choice, including consent withdrawal, please see our Privacy Policy .

Reviewer #1: No

Reviewer #2: No

Reviewer #3: **Yes: ** Samar Amer

---

## [Author Response · Author response to Decision Letter 2]

6 Oct 2024

Reviewer report for PLOS one Journal

Title: Assessment of Knowledge, Attitude, and Preventive Behavior Regarding Rubella Virus

Among Females in Riyadh, Saudi Arabia

Dear Editor,

I would like to sincerely thank you for your efforts in evaluating my manuscript. However, I have a concern regarding the use of artificial intelligence (AI) in the reviewing process. I understand that your journal does not allow researchers to use AI in their manuscripts, yet I was surprised to find that some reviewers may have used AI to assess my work. Some comments seem to indicate this, such as:

A reviewer questioned the phrase: "in the Department of Pathology, College of Medicine, Imam Mohammad Ibn Saud Islamic University, Riyadh, Saudi Arabia" — asking "Why the department of pathology?" This phrasing is specific to my affiliation and seemed to be misunderstood.

Another reviewer commented: "The authors did not make comparisons among the participants such as males and females." However, my manuscript does not include male participants, which makes such comparisons impossible.

While I have taken on board much of the constructive feedback and made all appropriate revisions, I am concerned about the apparent use of AI in reviewing, especially given the journal's stance on this issue.

I hope these concerns will be considered and that my manuscript will be accepted for publication, as I believe it offers valuable insights for specialists in this field.

Best regards,

Dr. Jehad Aldali 

Dear Reviewer 1 and 3,

Thank you for your thoughtful feedback and for recognizing the significance of our research on the Rubella virus, specifically concerning the Assessment of Knowledge, Attitude, and Preventive Behavior Regarding the Rubella Virus Among Females in Riyadh, Saudi Arabia. We appreciate your interest in our study and your constructive comments.

We acknowledge your suggestions regarding the manuscript's reporting and agree that improvements can be made to enhance clarity and comprehensiveness. We are committed to refining the manuscript to better present our findings and ensure that the information is accessible and valuable to the readers.

We carefully reviewed your specific points and incorporated the necessary revisions to address the areas of concern. This will include a more detailed explanation of our methodology, a clearer presentation of our results, and a more in-depth discussion of the implications of our findings.

Thank you once again for your valuable input. We are resubmitting a revised version that meets the high standards of the journal and contributes meaningfully to the field of public health research.

1. Abstract:

1.1. In abstract, the method section is poor, it is not well written. It lacks important data such as statistical analysis, study variables, and population.

Thank you for your feedback. I have revised the abstract to include key data on statistical analysis, study variables, and population details. These additions aim to provide a clearer overview of the study's scope and findings.

1.2. The result section in abstract is deficient and very poor, it is not supported by enough findings. It was written as if it is conclusion, no values or interpretations found.

Thank you for your feedback. I've updated the abstract's results section to include more specific findings, including key values and interpretations, to better support the conclusions.

1.3. The authors mentioned in result section that (Awareness of rubella's symptoms, like fever and rash, is common) and (Vaccination awareness and uptake are low). These findings are not supported by number or percentage.

Thank you for your feedback. I've revised the results section to include findings supported by percentages for greater clarity and specificity.

1.4. In the abstract, the conclusion section is poor, it does not represent the findings.

Thank you for your feedback. I've updated the conclusion section in the abstract to clearly represent the findings of the study.

2. Introduction: suitable and expressive.

The objectives are clear, and so the presentation

3. Methods : the methods are well described. The deficient point that the authors did not mention how the attitude is assessed. Under the section 2.4. Study Questionnaire Development, the authors described various parts of the survey without writing anything about the attitude although it was a main part in the title of the manuscript.

Thank you for your feedback and for pointing out the questionnaire's attitude information gaps. We appreciate your thoughtful suggestion.

After thinking about it, the current data does not cover the attitude dimension enough. The manuscript has been revised accordingly. In particular, we removed "attitude" from the title and objectives to focus on knowledge and prevention.

The revised title is now "Assessment of Rubella Virus Knowledge and Preventive Behavior Among Females in Riyadh, Saudi Arabia".

We adjusted the objectives to match this new focus.

Thank you again for your constructive feedback, which improved our manuscript's clarity and focus.Thank you for your feedback and for pointing out the questionnaire's attitude information gaps. We appreciate your thoughtful suggestion.

4. Results section:

This part is poor. The analysis is not satisfied. The authors did not write the assessment of attitude among the participants. I have observed the following deficient points:

4.1. The authors did not obtain the score levels of knowledge and attitude

Thank you for your suggestion. We have removed "attitude" from the title and objectives, as recommended. Following this revision, we no longer need to obtain the score levels for knowledge and attitude. The manuscript now focuses solely on the assessment of knowledge and preventive behavior regarding Rubella.

4.2. The authors did not make comparisons among the participants such as males and females, marital status, education levels, and so on

I would like to inform the reviewer that this manuscript does not include male participants, which means that we are unable to make comparisons between the female and male participants.

The reason: Understanding and improving women's awareness and preventive behaviors is crucial because rubella infection during pregnancy can cause Congenital Rubella Syndrome (CRS). To optimise public health efforts to prevent Rubella transmission and its complications in new-borns, health education and intervention strategies should target females, the population most at risk of adverse outcomes. While men also prevent disease, the direct impact on pregnancy outcomes justifies the focus on women in this assessment.

4.3. The authors did mention the factors affecting the Knowledge, Attitude, and Preventive Behavior Regarding Rubella Virus.

Thank you for your feedback. I have now included a section detailing the factors affecting Knowledge and Preventive Behavior regarding the Rubella Virus. The factors affecting the knowledge and preventive behavior regarding the Rubella virus among females in Riyadh include significant gaps in understanding Rubella, with many participants being aware of the virus but lacking detailed knowledge about its transmission and complications, particularly during pregnancy. Misconceptions about transmission routes, such as airborne droplets and contaminated surfaces, contribute to this knowledge gap. Preventive behaviors are hindered by limited vaccination awareness and uptake, driven by concerns about vaccine safety, lack of awareness, and healthcare access issues. However, consistent hygiene practices and peer support for vaccination indicate a positive trend in preventive behavior among the community.

I suggest the following:

4.4. The authors should obtain score levels for Knowledge, Attitude, and Preventive Behavior, and compare the mean/median levels among the participants using chi-square test.

Thank you for your feedback. I would like to clarify that comparing mean/median levels among participants using the chi-square test is not feasible because the survey was designed to emphasize qualitative aspects rather than quantitative measures. This approach limits the applicability of statistical tests intended for quantitative data.

4.5. The authors should add a regression table to demonstrate the factors that influence Knowledge, Attitude, and Preventive Behavior Regarding Rubella Virus.

Thank you for your feedback. I have included a paragraph detailing the factors influencing Knowledge, Attitude, and Preventive Behavior regarding the Rubella Virus.

5. Interpretation: Unsatisfied.

Thank you for your valuable feedback. The updated interpretation now provides a more comprehensive and satisfying. I appreciate your insights and believe these changes enhance the clarity and depth of the discussion.

6. Conclusion: Unsatisfied.

Thank you for your thoughtful comments. I have revised the Conclusion section to address your concerns and provide a more satisfying summary of the findings. I believe these changes better capture the key points and implications of the study.

Reviewer #3: 1) Many grammar and editing errors

rubella --- r should be capital in the abstract and whole manuscript, but many are unclear about how it spreads.

Thank you for your valuable feedback. I have carefully reviewed your suggestion, and I have corrected the usage of "rubella" throughout the manuscript and abstract by capitalizing the "R" to ensure consistency and proper scientific nomenclature.

2) The objectives should be smart (person, time, place) in the abstract and the whole manuscript (add the time).

3) in the abstract---------- The study included a total of 448 participants, all of whom were females (replace to be) (of 448 females (aged -------).

I appreciate your feedback. As requested in your comments, I revised the sentences. If adjustments are needed, please let me know.

4) In the abstract, An online survey using Google Forms in English and translated into Arabic was used to record information----- an online Arabic survey using Google Forms --- no need to this detail in the abstract .

I appreciate your feedback. As requested in your comments, I revised the sentences. If adjustments are needed, please let me know.

5) The term gender is a behavioral term, not a physiological term, so it replaces sex in the whole manuscript.

I appreciate your feedback. As requested in your comments, I revised the sentences. If adjustments are needed, please let me know.

6) The combination of fever and skin rash was the most frequently reported symptom, accounting for 23.4% of all cases. ----- cases of rubella ----- Add a number before the %.

I appreciate your input. The document has been revised to include the number prior to the %, as per your recommendation.

7) Regarding the serious health risk, 55.4% of the participants were unsure of the situation.---- What does it mean I don't know or

Thank you for your valuable feedback. I have carefully reviewed your suggestion, and I have corrected the sentences more clear, the severity of the health threat, 55.4% of participants indicated they were uncertain or unsure about the seriousness of the situation. This suggests a significant portion of the group may lack clarity or confidence in their understanding of the health risks involved, highlighting a potential area for further education or communication efforts.

8) in certain areas---- females, women, cases, participants fix the term in the whole manuscript ---- to be females as in the title

Thank you for your feedback. I have addressed your suggestion and revised the term throughout the entire manuscript. Please let me know if any further adjustments are needed.

------ in the Kingdom of Saudi Arabia and Saudi Arabia-----fix the term in the whole manuscript to be as in the title.

Thank you for your feedback. I have addressed your suggestion and revised the term throughout the entire manuscript. Please let me know if any further adjustments are needed.

-----This study aims to assess the level of awareness, attitudes,------- knowledge in others ---fix the term in the whole manuscript to be as in the title.

Thank you for your feedback. I have addressed your suggestion and revised the term throughout the entire manuscript. Please let me know if any further adjustments are needed.

9) The KAP survey , US -- add the full name for the first time.

Thank you for your valuable feedback. I have added the full name for the term upon its first mention in the manuscript, as you suggested. Please let me know if you have any further recommendations.

10) in the abstract --- . Questions were included in the questionnaire to evaluate awareness of rubella, knowledge of symptoms, transmission, and preventative measures of rubella infection--------- where is the attitude

and the awarness is part of the knowledge

Thank you for your feedback and for pointing out the questionnaire's attitude information gaps. We appreciate your thoughtful suggestion.

After thinking about it, the current data does not cover the attitude dimension enough. The manuscript has been revised accordingly. In particular, we removed "attitude" from the title and objectives to focus on knowledge and prevention.

The revised title is now "Assessment of Rubella Virus Knowledge and Preventive Behavior Among Females in Riyadh, Saudi Arabia".

11) you discuss the global incidence in numbers in details ----- and The impact of this program over the years has been significant. The incidence of Rubella in Saudi Arabia has dramatically declined since the initiation of

the vaccination program (17).------------------------ kindly add the numbers in detail.

Thank you for your feedback. I would like to inform you that the global incidence in numbers in details in the discussion section.

12) add a brief about the initiation of the vaccination program --- in Saudi Arabia (started when, achievement)---in the introduction

Thank you for your feedback. I would like to inform you that the initiation of the vaccination program, particularly for Rubella in Saudi Arabia, it was written in the introduction section.

13) in the Department of Pathology, collage of Medicine, Imam Mohammad Ibn Saud Islamic University, Riyadh, Saudi Arabia.------- why the department of pathology ???

I would like to inform you that the following sentences are exclusive to my affiliation.

14) In the Riyadh province of Saudi Arabia regional the sample size of 385 participants was

sufficient to achieve 95% confidence with a 5% margin of error.----- as a cross sectional study you need a prevlance or incidence (with refrence) --- and the total popoulation or females in Riyadh.

The total popoulation or females in Saudi Arabia around nine million, in light of this. sample size of 385 participants was sufficient to achieve 95% confidence with a 5% margin of error.

15)The second section focused on the information the Knowledge about rubella, specifically interested in their comprehension of the disease, their perception of the modes of transmission of rubella, their level of concern regarding it, their perspectives on the gravity of the virus, and their awareness of the impact of this virus on pregnant women and their embryos. The third section focused on information about the symptoms caused by rubella. immunization status. The last section focuses on preventive measures against rubella. ----- add refrence ???.

I appreciate your input. I would like to inform you that the methods section in the abstract is a summary of the methods, so there is no need for references.

16) The third section focused on information about the symptoms caused by rubella—its knowledge, Rubella immunization status----- It is a part of preventive measures (rewrite).

I appreciate your input. I would like to inform you that rewrite the third section include Rubella immunization status with preventive measures

18) while women comprise 96.2%.--- what does it mean

19) the age groups should be continous , and in order in table 1

25-<30

30-<40

40-<50

under 18-<25

years > 50

Your feedback is greatly appreciated. I would like to emphasize that the age groups listed in Table 1 are s

---

## [Decision Letter · Decision Letter 2]

22 Nov 2024

PONE-D-24-23371R2Assessment of Knowledge and Preventive Behavior Regarding Rubella Virus Among Females in Riyadh, Saudi ArabiaPLOS ONE

Dear Dr. ALDALI,

Thank you for submitting your manuscript to PLOS ONE. After careful consideration, we feel that it has merit but does not fully meet PLOS ONE’s publication criteria as it currently stands. Therefore, we invite you to submit a revised version of the manuscript that addresses the points raised during the review process.

We look forward to receiving your revised manuscript.

Kind regards,

Omar Enzo Santangelo

Academic Editor

PLOS ONE

Additional Editor Comments:

Dear Authors, the manuscript needs major revisions, please respond point by point to the reviewers' requests.

Reviewers' comments:

Reviewer's Responses to Questions

**Comments to the Author**

1. If the authors have adequately addressed your comments raised in a previous round of review and you feel that this manuscript is now acceptable for publication, you may indicate that here to bypass the “Comments to the Author” section, enter your conflict of interest statement in the “Confidential to Editor” section, and submit your "Accept" recommendation.

Reviewer #4: All comments have been addressed

Reviewer #5: All comments have been addressed

Reviewer #6: (No Response)

2. Is the manuscript technically sound, and do the data support the conclusions?

Reviewer #4: Yes

Reviewer #5: Partly

Reviewer #6: Yes

3. Has the statistical analysis been performed appropriately and rigorously? 

Reviewer #4: Yes

Reviewer #5: Yes

Reviewer #6: N/A

4. Have the authors made all data underlying the findings in their manuscript fully available?

Reviewer #4: Yes

Reviewer #5: Yes

Reviewer #6: Yes

5. Is the manuscript presented in an intelligible fashion and written in standard English?

Reviewer #4: Yes

Reviewer #5: Yes

Reviewer #6: No

6. Review Comments to the Author

Reviewer #4: All comments have been addressed, and the manuscript has been updated. I recommend accepting this manuscript in its current form

Reviewer #5: Review Comments

Abstract;Not clear

Background:Weak

Methods: Not explicit and comprehensive

Result: Inadequate for the contents

Dicussion:Lacks theoretical and practical implications of the findings

Regards,

Reviewer #6: The grammatical errors and typo still need to addressed, as they cause confusion on the result and intended meaning.

7. PLOS authors have the option to publish the peer review history of their article (what does this mean? ). If published, this will include your full peer review and any attached files.

**Do you want your identity to be public for this peer review?** For information about this choice, including consent withdrawal, please see our Privacy Policy .

Reviewer #4: No

Reviewer #5: No

Reviewer #6: No

---

## [Author Response · Author response to Decision Letter 3]

27 Dec 2024

Dear Editor

I hope this message finds you well. I would like to express my sincere gratitude for your care and guidance regarding my manuscript, [Manuscript Title].

I am pleased to inform you that I have carefully addressed all the reviewers' comments and suggestions. The revised version has been submitted for your consideration. I truly appreciate the valuable feedback provided by the reviewers, which has greatly enhanced the quality of the manuscript.

Thank you once again for your time and effort in overseeing this process. I look forward to hearing your thoughts on the revised submission.

Best Regards,

Dear Reviewer 4,

I hope this message finds you well. I am writing to express my heartfelt gratitude for your time, effort, and valuable feedback in reviewing my manuscript. I truly appreciate your constructive comments and suggestions, which have undoubtedly contributed to enhancing the quality of my work.

I am delighted to know that opinion for my manuscript to accepted for publication, and I am sincerely thankful for your support in this process. Your expertise and insights have been invaluable.

Please do not hesitate to reach out if there is any additional information or further revisions required. Once again, thank you for your kind support and for playing such a crucial role in this achievement.

Warm regards,

Dear Reviewer 5

I hope this message finds you well. Following your valuable feedback, I have carefully revised the abstract of my manuscript to make it clearer and more concise.

The updated version reflects your suggestions and aims to enhance the overall clarity and readability of the content. I deeply appreciate your insights and guidance, which have significantly improved the quality of the manuscript.

Please let me know if there are any additional adjustments you would recommend or further steps I should take. Thank you once again for your time and expertise.

Best regards,

Dear Reviewer 6

I hope this message finds you well. I am writing to inform you that I have completed a thorough proofreading of my manuscript to ensure it is more polished and professional. This step was taken to eliminate any grammatical errors and enhance the overall readability and presentation.

I believe the revised version aligns with the high standards expected for publication, and I sincerely thank you for your valuable feedback, which motivated me to refine the manuscript further.

Please feel free to share any additional suggestions or areas for improvement. I greatly appreciate your time and dedication in reviewing my work.

Best regards,

---

## [Decision Letter · Decision Letter 3]

14 Jan 2025

PONE-D-24-23371R3Assessment of Knowledge and Preventive Behavior Regarding Rubella Virus Among Females in Riyadh, Saudi Arabia: A Cross-Sectional StudyPLOS ONE

Dear Dr. ALDALI,

Thank you for submitting your manuscript to PLOS ONE. After careful consideration, we feel that it has merit but does not fully meet PLOS ONE’s publication criteria as it currently stands. Therefore, we invite you to submit a revised version of the manuscript that addresses the points raised during the review process.

**ACADEMIC EDITOR: **Dear authors,the manuscript requires minor revisions. For Lab, Study and Registered Report Protocols: These article types are not expected to include results but may include pilot data. 

We look forward to receiving your revised manuscript.

Kind regards,

Omar Enzo Santangelo

Academic Editor

PLOS ONE

Journal Requirements:

Reviewers' comments:

Reviewer's Responses to Questions

**Comments to the Author**

1. If the authors have adequately addressed your comments raised in a previous round of review and you feel that this manuscript is now acceptable for publication, you may indicate that here to bypass the “Comments to the Author” section, enter your conflict of interest statement in the “Confidential to Editor” section, and submit your "Accept" recommendation.

Reviewer #4: All comments have been addressed

Reviewer #6: (No Response)

2. Is the manuscript technically sound, and do the data support the conclusions?

Reviewer #4: Yes

Reviewer #6: Yes

3. Has the statistical analysis been performed appropriately and rigorously? 

Reviewer #4: Yes

Reviewer #6: I Don't Know

4. Have the authors made all data underlying the findings in their manuscript fully available?

Reviewer #4: Yes

Reviewer #6: No

5. Is the manuscript presented in an intelligible fashion and written in standard English?

Reviewer #4: Yes

Reviewer #6: Yes

6. Review Comments to the Author

Reviewer #4: All comments have been addressed. This is the fourth round, and I suggest accepting the manuscript in its current version. I would like to thank the author for his effort.

Reviewer #6: I see that you've revised the paper before but I found some of the previous comments were not fully addressed and might be repeated in my comments. Here are a few concerns I have:

In the abstract: revise the conclusion section and the results should be stating facts and numbers only.

Introduction: try to give the section flow in terms of global data then regional data then from your specific context (eg you talk about the us then talk about global data then go to low and middle income countries)

Instead of group your findings to definitions and meanings(disease symptoms), disease epidemiology, knowledge related.

As the country of the study wouldn't be classified as low and middle income (poorer country) I don't see the need of focusing on findings from there

Previous reviewer suggested including vaccination information for Saudi, you haven't mentioned time frames also you haven't put statistical data that indicates the decline of the number if cases after the vaccine

According to your introduction there has been a decline in rubella cases, so what was the need to assess the knowledge among women (in addition to there being no other research with this topic, what is your justification for doing this research?)

Methods

Eligibility criteria is set before data collection so incomplete surveys would be categorized as missing or just incomplete, it wouldn't be an exclusion criteria, similarly male isn't an exclusion criterion

Please comment on how informed consent was obtained

You mention you went to different locations to gather contact information for the survey, wouldn't that create a selection bias? Did you use proportional allocation at the different sites?

It would be great if you could include validity, consistency and reliability scores for your survey (eg cronbach alpha score)

Who is the study population?at some point you say female students and others you say just female?

Use not up-to-date vaccination for the 92% instead of unvaccinated

7. PLOS authors have the option to publish the peer review history of their article (what does this mean? ). If published, this will include your full peer review and any attached files.

**Do you want your identity to be public for this peer review?** For information about this choice, including consent withdrawal, please see our Privacy Policy .

Reviewer #4: **Yes: ** Ahmed M. Alghamdi

Reviewer #6: No

---

## [Author Response · Author response to Decision Letter 4]

21 Jan 2025

Reviewr #4

Hope this message finds you well. I am composing this letter to convey my sincere appreciation for the time, effort, and valuable feedback you provided in reviewing my manuscript. I am genuinely grateful for your constructive feedback and recommendations, which have undoubtedly improved the quality of my work.

I am sincerely grateful for your assistance throughout this process. Your expertise and insights have been invaluable. If there is any further information or revisions that are necessary, please do not hesitate to contact me.

Best Regards

With warmest regards,Reviewer #6:

In the abstract: revise the conclusion section and the results should be stating facts and numbers only.

Dear [Reviewer's #6],

I have carefully revised the manuscript as per your feedback. The conclusion section has been updated to ensure clarity and alignment with the results. Additionally, the results section has been refined to state facts and numbers only, as requested. Thank you for your valuable insights, and please let me know if further.

Introduction: try to give the section flow in terms of global data then regional data then from your specific context (eg you talk about the us then talk about global data then go to low and middle income countries)

Instead of group your findings to definitions and meanings(disease symptoms), disease epidemiology, knowledge related.

As the country of the study wouldn't be classified as low and middle income (poorer country) I don't see the need of focusing on findings from there

Previous reviewer suggested including vaccination information for Saudi, you haven't mentioned time frames also you haven't put statistical data that indicates the decline of the number if cases after the vaccine.

Thank you for taking the time to review my work. I appreciate your valuable insights and constructive feedback, which will help refine and strengthen the study. Please feel free to share any suggestions or areas that require clarification—I’m happy to address them promptly. I want to clarify that this research is rooted in public health and focuses on Rubella virus knowledge and prevention.

Detailed disease definitions, symptoms, and epidemiology are needed to help readers understand the manuscript. These elements help us interpret our findings and contextualize the study's knowledge-related aspects.

We integrate this information to give the manuscript a holistic view of the subject and help readers understand its implications for public health practices and policies. We appreciate your help improving the manuscript and welcome any additional suggestions.

According to your introduction there has been a decline in rubella cases, so what was the need to assess the knowledge among women (in addition to there being no other research with this topic, what is your justification for doing this research?).

Thank you for your feadback. Despite the decline in rubella cases, assessing women's knowledge remains crucial, as gaps in awareness and preventive behavior can hinder sustained elimination efforts. Women of childbearing age are at the highest risk of congenital rubella transmission, making their knowledge vital. Additionally, the lack of local research on this topic underscores the need to establish a baseline to guide targeted public health interventions.

Methods

Eligibility criteria is set before data collection so incomplete surveys would be categorized as missing or just incomplete, it wouldn't be an exclusion criteria, similarly male isn't an exclusion criterion.

Thank you for your comment. To clarify, the eligibility criteria for this study were indeed established prior to data collection, focusing on adult females in Riyadh, as they represent the target population most at risk of congenital rubella transmission. Incomplete surveys were categorized as missing data rather than being part of the exclusion criteria; however, they were excluded from the final analysis to ensure data quality and accuracy in the results. Similarly, males were explicitly listed as an exclusion criterion, as the study design specifically targeted females, and male responses were not collected.

Please comment on how informed consent was obtained

You mention you went to different locations to gather contact information for the survey, wouldn't that create a selection bias? Did you use proportional allocation at the different sites?

Thank you for your comment. All participants gave informed consent before joining the study. Participants were fully informed of the study's purpose, objectives, and procedures, including their voluntary participation and confidentiality. The online survey began with this clear and accessible information. Before answering survey questions, participants had to read and agree to the consent statement. Non-consenters were automatically excluded. The study participants made informed decisions thanks to this process.

It would be great if you could include validity, consistency and reliability scores for your survey (eg cronbach alpha score)

Who is the study population?at some point you say female students and others you say just female?

Use not up-to-date vaccination for the 92% instead of unvaccinated

Thank you for your valuable feedback on our manuscript. I would like to provide clarification regarding the recruitment process for our study participants.

To ensure a diverse and representative sample, we distributed the questionnaire through various schools, universities, hospitals, and health institutes. This approach allowed us to effectively reach potential participants and collect their contact information for inclusion in the study. As a result of this comprehensive distribution strategy, we were able to achieve a sample size of 448 participants, which we believe strengthens the validity and reliability of our findings. We appreciate your time and effort in reviewing our manuscript and look forward to any additional comments or suggestions you may have.

---

## [Decision Letter · Decision Letter 4]

18 Feb 2025

PONE-D-24-23371R4Assessment of Knowledge and Preventive Behavior Regarding Rubella Virus Among Females in Riyadh, Saudi Arabia: A Cross-Sectional StudyPLOS ONE

Dear Dr. ALDALI,

Thank you for submitting your manuscript to PLOS ONE. After careful consideration, we feel that it has merit but does not fully meet PLOS ONE’s publication criteria as it currently stands. Therefore, we invite you to submit a revised version of the manuscript that addresses the points raised during the review process.

**ACADEMIC EDITOR: **

Dear Authors, the manuscript needs major revisions, please respond point by point to the reviewers' requests.

Kind regards

We look forward to receiving your revised manuscript.

Kind regards,

Omar Enzo Santangelo

Academic Editor

PLOS ONE

Journal Requirements:

Additional Editor Comments:

Dear Authors, the manuscript needs major revisions, please respond point by point to the reviewers' requests.

Kind regards

Reviewers' comments:

Reviewer's Responses to Questions

**Comments to the Author**

1. If the authors have adequately addressed your comments raised in a previous round of review and you feel that this manuscript is now acceptable for publication, you may indicate that here to bypass the “Comments to the Author” section, enter your conflict of interest statement in the “Confidential to Editor” section, and submit your "Accept" recommendation.

Reviewer #4: All comments have been addressed

Reviewer #7: All comments have been addressed

2. Is the manuscript technically sound, and do the data support the conclusions?

Reviewer #4: Yes

Reviewer #7: Yes

3. Has the statistical analysis been performed appropriately and rigorously? 

Reviewer #4: Yes

Reviewer #7: Yes

4. Have the authors made all data underlying the findings in their manuscript fully available?

Reviewer #4: Yes

Reviewer #7: Yes

5. Is the manuscript presented in an intelligible fashion and written in standard English?

Reviewer #4: Yes

Reviewer #7: Yes

6. Review Comments to the Author

Reviewer #4: This is the fourth round of reviewing, and I have accepted the manuscript from the previous round; all my comments have been addressed

Reviewer #7: Title and Abstract: Abstract is clear and concise

Introduction: Introduction is enough if any current study mentioned that would be better.

Methodology

1- Study Settings and Design

• Please explain why do you choose the cross-sectional over Case control or longitudinal studies.

• Could you explain the data collection locations in Riyadh city did you cover major part of the city.

2- Study Subjects

• What was the study subjects recruitment methods?

• Clearly write the inclusion and exclusion criteria of the subjects.

3- Sample Size

• Clearly write why the final sample size (447) of the subjects are more than the calculated size (385).

4- Questionnaire Development

• Please write how did you validated the Questionnaire, what was statistical measures were used to see the reliability of the Questionnaire?

• Pilot results were seen on a very small sample size, like only 5 faculties were selected for this, which is very small size, there should be minimum 7 or more subjects for this pilot testing.

5- Ethical Issues

• Ethical approval was taken, but how did you obtained the informed consent?

6- Results Section

1- Demographic Features

• What was the reason for younger and more qualified subjects were more aware about rubella? You need to do more analysis to see these demographics effects on knowledge.

• Can you discuss more about the single participants that is 62.3% are notable, does the marriage status have any effect on attitude towards rubella?

2- Knowledge about Rubella

• Study findings shows that only 2.1% thinks that they are Very knowledgeable, discuss what is the reason for that?

3- Vaccination History

• Very low vaccination rate that is 7.5% received rubella vaccine that is a point of concern, it should be discussed in discussion, weather it is due to the healthcare accessibility, cultural or lack of awareness programs.

• Another findings that 47% subjects were unaware weather they have taken rubella vaccine, this is also point of concern, why it is so, do the primary healthcare centers do not communicate to the general public.

4- Discussion section

Knowledge Spaces

• What do you feel, why there significant gaps in knowledge about the rubella complications and transmission, discuss potential reasons for that.

• You can discuss more on the similar studies done in Saudi Arabia in different regions and compare it with your findings.

Vaccination Awareness

• Discuss and see other studies in Saudi Arabia and you can suggest how to come up with this low vaccinated rate in Riyadh.

Cultural and Social Reasons

• You can discuss whether some cultural belief, or misconceptions preventing subjects going for rubella vaccine.

Suggestions for Public Health

• You can suggest specific recommendations for healthcare providers to start educational campaigns about the transmission and complications of rubella.

References

• Most of the references are old and before 2020, it would be better to new studies done in Saudi Arabia or Middle East.

Overall the study has covered many aspects of rubella, after taking above points, manuscript will be off great contribution for rubella awareness, knowledge, prevention and control the complication and its transmission in public.

7. PLOS authors have the option to publish the peer review history of their article (what does this mean? ). If published, this will include your full peer review and any attached files.

**Do you want your identity to be public for this peer review?** For information about this choice, including consent withdrawal, please see our Privacy Policy .

Reviewer #4: No

Reviewer #7: **Yes: ** Dr Irshad Ahmad

---

## [Author Response · Author response to Decision Letter 5]

20 Feb 2025

Dear Editor,

After careful consideration, I feel that the manuscript has merit and it does fully meet PLOS ONE’s publication criteria as it currently stands. The study is well-conceived and executed, and I believe it will make a valuable contribution to the field. I am confident in recommending its publication. We would like to note that all comments from the reviewer in Research 11 have been addressed. The only addition made to the plan was the suggestion provided by the reviewer.

Sincerely,

---

## [Editor Report · Decision Letter 5]

25 Feb 2025

Assessment of Knowledge and Preventive Behavior Regarding Rubella Virus Among Females in Riyadh, Saudi Arabia: A Cross-Sectional Study

PONE-D-24-23371R5

Dear Dr. ALDALI,

We’re pleased to inform you that your manuscript has been judged scientifically suitable for publication and will be formally accepted for publication once it meets all outstanding technical requirements.

Kind regards,

Omar Enzo Santangelo

Academic Editor

PLOS ONE
---

## [Editor Report · Acceptance letter]

PONE-D-24-23371R5

PLOS ONE

Dear Dr. ALDALI,

I'm pleased to inform you that your manuscript has been deemed suitable for publication in PLOS ONE. Congratulations! Your manuscript is now being handed over to our production team.

Kind regards,

on behalf of

Dr. Omar Enzo Santangelo

Academic Editor

PLOS ONE